# Incidence of Congenital Clubfoot: Preliminary Data from Italian CeDAP Registry

**DOI:** 10.3390/ijerph19095406

**Published:** 2022-04-29

**Authors:** Daniela Dibello, Lucio Torelli, Valentina Di Carlo, Adamo Pio d’Adamo, Flavio Faletra, Alessandro Mangogna, Giulia Colin

**Affiliations:** 1Unit of Paediatric Orthopaedics and Traumatology, Giovanni XXIII Children’s Hospital, Via Giovanni Amendola, 70126 Bari, Italy; daniela.dibello@policlinico.ba.it; 2Clinical Department of Medical, Surgical and Health Sciences University of Trieste, Strada di Fiume, 34149 Trieste, Italy; torelli@units.it (L.T.); adamopio.dadamo@burlo.trieste.it (A.P.d.); 3Unit of Paediatric Orthopaedics and Traumatology, Institute for Maternal and Child Health—IRCCS Burlo Garofolo, Via dell’Istria, 34137 Trieste, Italy; 4Medical Genetics Unit, Institute for Maternal and Child Health—IRCCS Burlo Garofolo, Via dell’Istria, 34137 Trieste, Italy; flavio.faletra@burlo.trieste.it (F.F.); alessandro.mangogna@burlo.trieste.it (A.M.); 5Unit of Orthopaedics and Traumatology, University of Trieste, Strada di Fiume, 34149 Trieste, Italy; giulia.colin@burlo.trieste.it

**Keywords:** congenital malformation, epidemiology, newborn, clubfoot, congenital talipes equinovarus

## Abstract

(1) Background: We find the incidence of clubfoot in Italy from “Certificate of Delivery Care Registry (CeDAP)”, a database of the Italian Ministry of Health, the most comprehensive public data available for this purpose. (2) Methods: The CeDAP registry is a web system that provides epidemiological and sociodemographic information about newborns. It started on 1 January 2002, following the ministerial Decree no. 349 of 16 July 2001. The certificate is structured into six sections; each collects specific information referring to the birthplace, parents, pregnancy, childbirth, newborn, and the possible presence of congenital malformations or the causes of neonatal mortality. The midwife or the doctor draws up the certificate no later than ten days after birth. Each region transmits the data every six months to the Ministry of Health. The period between 2013 and 2017 has been selected for the study, with every Italian region’s data. We conducted a retrospective descriptive study. (3) Results: The overall rate in northern Italy is 1.09 (with some exceptions described), but we think it is essential to reevaluate this number again, given more accurate data collections by every Italian hospital. (4) Conclusions: This study intends to build a framework for future epidemiologic studies about clubfoot in Italy.

## 1. Introduction

Congenital clubfoot, best described as congenital talipes equinovarus (CTEV), is the most frequent foot congenital deformity. CTEV is one of the most common pediatric malformations, and it consists of a heterogeneous group of abnormalities that give a permanent deviation of the foot and ankle if untreated [1]. This pediatric defect is clinically classified as secondary or isolated; it is secondary or syndromic when associated with another congenital disease (20% of cases), and isolated if no other malformations are present (80% of cases), which introduces the concept of idiopathic CTEV [1,2]. The congenital defect differs from postural clubfoot because it is flexible to clinical maneuvers due to its prenatal positional nature and it is easily treatable [3,4]. Postural clubfoot is a structurally normal foot and, despite an abnormal resting position, the doctor can correct it with specific physical manipulations [3]. However, children affected by CTEV, if they were not treated adequately, will face more difficulties in everyday life, as they do not have a standard anatomical and functional foot [3,5].

The exact etiology of CTEV has not yet been identified, but it involves both genetics and environmental factors [6]. Indeed, although it has a more remarkable recurrence within families, a twin study has suggested a significant role of environmental factors in etiology [6]. In particular, maternal smoking [1,7], obesity [8], and the use of selective serotonin reuptake inhibitors [9] have been related to the risk of CTEV. 

Prenatal diagnosis is possible with an ultrasound routine check at around 20 weeks, which can recognize the different forms of clubfoot. Prenatal ultrasound has an accuracy of 86% for isolated clubfoot and usually correlates with postnatal severity with a higher Pirani score [10]. The prenatal diagnosis can help and psychologically prepare the parents for the pathology and the treatment [4,11]. Over the years, various treatments with different degrees of surgical invasiveness have been proposed [12]. The clubfoot surgical approach was commonly associated with complications, with the clinical outcome of a foot unable to achieve complete function due to subsequent retractions and scarring. The Ponseti method is recognized worldwide and supported by the World Health Organization as the gold standard in clubfoot treatment [13]. The method is a quick, cost-effective, and painless treatment that can start few days after birth. The treatment consists of a series of manipulations followed by a series of 5–6 casts to maintain the correct position of the foot. Every cast must stay in place for 5–7 days for the ligamentous structures to adapt to the new position. The aim of the cast phase is to correct the cavism, adduction, internal torsion, and varism. The last deformity to be corrected is equinism. The technique involves a day surgery percutaneous Achilles tenotomy (20–30% of all cases) to fix the residual equinism, followed by a cast for 20 days to help healing the tendon in elongation. The last step of the treatment requires using the Mitchell Ponseti brace up to 5 years of age with a dedicated daily routine that is gradually reduced during the years [14].

It is estimated that 0.5–2 of every 1000 newborns are affected (150,000–200,000 newborns per year and 7–43 cases of clubfoot/year/million population), with a male to female ratio of 2:1 and major distribution in developing countries (80%) [12,13]. Kruse and colleagues suggested a reason for this gender difference in the Carter effect [14]. In 50% of cases, it affects both feet and the right side more than the left one in unilateral clubfoot [15]. According to some epidemiological investigations, major differences in prevalence have been identified between several race-ethnic groups, reaching the higher rates in Maori (7/1000 newborns) [16], in Polynesians and Hawaiians (6.8/1000 newborns) [17], and in the population of southern Africa (3.5/1000 newborns) [18]. Whereas, the percentages in the Chinese population (0.39/1000 newborns) [2], in the Japanese (0.87/1000 newborns) [19], in the Asian (0.57/1000 newborns), in European (1.2/1000 newborns) [20], and in the Brazilian population (1.7/1000 newborns) [21] have been lower. In the latest Italian regional report, Pavone and collaborators analyzed the Sicilian population from 1991 to 2004 and observed 827 cases of ICTEV out of 801,324 live births, with a prevalence of about 1:1000, a male to female ratio of 2:1, and the right foot affected slightly more frequently than the left [22]. Recently, it has been published the “European Surveillance of Congenital Anomalies (EUROCAT)” concerning the clubfoot prevalence in newborns, but the data regarding Italy are missing or poor, including only the regions of Tuscany and Emilia Romagna [23]. EUROCAT is the European network of congenital anomaly registers, covering about 30% of the European birth population (EU and non-EU countries) [24]. 

This paper aims to fill this gap by publishing the data taken from the CEDAP registry of the Italian Ministry of Health. As far as we know, these are the most comprehensive public data available for clubfoot prevalence in Italy.

## 2. Materials and Methods

The “Certificate of Delivery Care Registry (CeDAP)” is a web system that provides epidemiological and sociodemographic information about newborns, essential for Italian public health and health statistics. The current data collection of the CeDAP started on 1 January 2002, following the ministerial Decree no. 349 of 16 July 2001. The certificate is structured into six sections; each collects specific information referring to the birthplace, parents, pregnancy, childbirth, newborns, and the possible presence of congenital malformations or the causes of neonatal mortality. The certificate is drawn up no later than the tenth day after birth by the midwife or the doctor who assisted. In case of dead-births or fetal malformations, specific information is collected in the certificate. Each region transmits the data every six months to the Ministry of Health according to the following timing:-By September 30, the data relating to the first half of the year considered;-By March 31, the data concerning the second half of the previous year, and any corrections and additions related to the first half [25].

The period between 2013 and 2017 has been selected for the study, collecting every Italian region’s data.

We decided to use the CeDAP registry, although there are other quality reporting systems such as EUROCAT, an extremely valid method for the epidemiological surveillance of congenital anomalies, which was not adopted because not all the Italian regions have provided complete data on newborns affected by clubfoot. 

We conducted a retrospective descriptive study. 

## 3. Results and Discussion

Between 2013 and 2017, 1379 alive newborns with clubfoot were reported by the CeDAP registry, giving Italy an overall prevalence rate of 0.57 every 1000 newborns. This number differs from European data, where the prevalence appears to be 1.13 per 1000 births [23]. We noticed an increasing prevalence of the pathology in the considered period, despite some missing regional data as shown in Table 1 and Table 2.

Notably, we have no data for a region with few inhabitants such as Basilicata and a very populous region such as Lazio. There are also regions and independent provinces that did provide incomplete information about the period described.

From the data reported, it can be seen that Lombardia (10.02 million inhabitants in 2017) (Eurostat data), a populous region, has more or less the identical prevalence rate of the smallest Valle d’Aosta or Provincia Autonoma di Bolzano.

If we analyze only the regions that gave their numbers every year, expecting them to be complete, the numbers are slightly different. As we can notice, studying the data in Table 3, the overall weighted average is 1.09 every 1000 newborns instead of 0.57 (Table 2), which is closer to the European rate of 1.13 [23]. We have no information about differentiation on isolated congenital clubfoot nor syndromic ones or severity of the anomaly.

Regarding the missing data, we can speculate.

It can be assumed that some regions do not have orthopedic-pediatric referral centers for clubfoot treatment, and in this case, the children and their families may have moved from one region to another. The same could have happened if the defects were diagnosed during pregnancy, leading the mother to choose another area with a referral center to be followed and give birth.

In our opinion, these inconsistencies are also due to the variability of data collection accuracy in Italy, since each region has different autonomy in the health system, and there are no national standards for data collection. 

The limit of our study is that the data obtained from the Ministry of Health are incomplete and a statistical analysis is not possible. The main aim of our work is to present and describe, for the first time, available Italian data. Our hope is that our effort could be the basis for further studies.

As far as we are concerned, we would like to start an inter-regional collaboration to find regional coordinators who will collect hospital data about clubfoot. Our effort aims to start a prospective multicentric study to describe the official incidence rate of clubfoot in Italy.

## 4. Conclusions

It is estimated that from every 1000 newborns, 1–2 newborns are affected by congenital clubfoot worldwide [12]. Italian official data are missing; hence, there is the need to overcome the matter with CeDAP Registry records. We found out that the overall rate in northern Italy is 1.09 (with the exceptions described). These are preliminary data and we believe that the request for better centralization of official data or more accurate data collections by each Italian hospital is essential. For this reason, our effort for the next years will be to start a prospective study to analyze and describe the incidence of clubfoot in Italy. This study is a framework for future epidemiologic studies in Italy and a starting point for genetic investigations related to the epidemiology of individual Italian regions or clusters.

## Figures and Tables

**Table 1 ijerph-19-05406-t001:** Newborns with clubfoot.

Italian Regions	2013	2014	2015	2016	2017	Total 2013–2017
Piemonte	39	27	28	43	33	170
Valle d’Aosta	3	4	2	1	2	12
Lombardia	66	97	144	160	184	651
Provincia Autonoma Bolzano	9	13	11	14	11	58
Provincia Autonoma Trento	6					6
Veneto	35	26	26	26	34	147
Friuli-Venezia Giulia	5	9	13	6	3	36
Liguria	1	6	4	1	2	14
Emilia-Romagna	19	31	21	18	14	103
Toscana	5	4	5	8	9	31
Umbria		1	1		2	4
Marche	4	9	1	2	2	18
Lazio						0
Abruzzo	3	1	1	1	1	7
Molise	2				1	3
Campania		2	1			3
Puglia	7	5	2	3	6	23
Basilicata						0
Calabria	7	5	6	9	8	35
Sicilia	5	14	7	14	16	56
Sardegna			1		1	2
**Total**	**216**	**254**	**274**	**306**	**329**	**1379**

**Table 2 ijerph-19-05406-t002:** Rate per 1000 newborns.

Italian Regions	2013	2014	2015	2016	2017	Total 2013–2017
Piemonte	1.16	0.83	0.90	1.36	1.07	1.06
Valle d’Aosta	2.63	3.48	2.06	1.04	2.21	2.34
Lombardia	0.74	1.11	1.69	1.96	2.33	1.54
Provincia Autonoma Bolzano	1.62	2.28	2.01	2.57	2.06	2.11
Provincia Autonoma Trento	1.27	0.00	0.00	0.00	0.00	0.26
Veneto	0.84	0.65	0.67	0.69	0.93	0.75
Friuli-Venezia Giulia	0.53	0.97	1.47	0.71	0.37	0.81
Liguria	0.09	0.57	0.40	0.10	0.21	0.28
Emilia-Romagna	0.50	0.84	0.59	0.52	0.42	0.58
Toscana	0.17	0.14	0.18	0.30	0.34	0.22
Umbria	0.00	0.13	0.14	0.00	0.33	0.12
Marche	0.32	0.74	0.09	0.17	0.19	0.31
Lazio						
Abruzzo	0.29	0.10	0.10	0.10	0.11	0.14
Molise	1.11	0.00	0.00	0.00	0.47	0.30
Campania	0.00	0.04	0.02	0.00	0.00	0.01
Puglia	0.20	0.15	0.06	0.10	0.20	0.14
Basilicata						
Calabria	0.43	0.31	0.38	0.56	0.51	0,44
Sicilia	0.11	0.31	0.16	0.34	0.39	0.26
Sardegna	0.00	0.00	0.09	0.00	0.10	0.04
**Total**	**0.42**	**0.51**	**0.56**	**0.65**	**0.72**	**0.57**

**Table 3 ijerph-19-05406-t003:** Clubfoot incidence rate about regions who gave the data.

Italian Regions	2013	2014	2015	2016	2017	Total 2013–2017
Piemonte	1.16	0.83	0.90	1.36	1.07	1.06
Valle d’Aosta	2.63	3.48	2.06	1.04	2.21	2.34
Lombardia	0.74	1.11	1.69	1.96	2.33	1.54
Provincia Autonoma Bolzano	1.62	2.28	2.01	2.57	2.06	2.11
Veneto	0.84	0.65	0.67	0.69	0.93	0.75
Friuli-Venezia Giulia	0.53	0.97	1.47	0.71	0.37	0.81
Liguria	0.09	0.57	0.40	0.10	0.21	0.28
Emilia-Romagna	0.50	0.84	0.59	0.52	0.42	0.58

## Data Availability

Data Availability Statements at https://www.salute.gov.it/portale/home.html (accessed on 13 March 2021).

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
