# Peer review of "Incidence of Congenital Clubfoot: Preliminary Data from Italian CeDAP Registry"

_ijerph, 2022, doi:10.3390/ijerph19095406_

Round 1

Reviewer 1 Report

Dear authors and editor

I would like to justify why I do not recommend that this work be published in its present form.

ABSTRACT:

Background: Describing CeDAP is not the background of the study for me.

Keywords: I suggest adding two words:  clubfoot and congenital talipes equinovarus.

Introduction

Line 41. All your cases present congenital clubfoot deformity.  Fewer of them have genetical implications and other children have postural etiopathogenesis. So, the sentence: “The congenital defect differs from postural clubfoot because it is flexible” is very confusing, because both circumstances are detected at the birth moment so they are both considered congenital.

Would you mean “other congenital defects as metatarsus varus? 

Materials and Methods

Line 117. Were all newborns, 1379, alives?

Results and discussion.

The results are a simple description of data with little explications. Deficient statistical treatment is performed nor are other related variables considered.

Discussion is very poor.

The results are a simple description of data with little explanation. A poor statistical treatment is performed and other related variables are not considered.

The discussion is very poor.

Author Response

Dear Reviewer,

Thank you for your tips. We made corrections as suggested.

The data presented in the manuscript are incomplete because some Italian regions did not gave it.

The aim of the work is to frame the Italian data available about clubfoot, that are incomplete as we did explain in the manuscript.

As we underline in the paper, statistical analysis is not possible with incomplete data.

Reviewer 2 Report

An interesting work, a subject that has not been fully explored and fully explained so far. I have comments: Introduction: Please add that 5-6 casts are applied in the Ponseti method, Achilles tenotomy is performed only in 20-30% of patients, and the internal torsion of the foot is also corrected. Materials and Methods: Clubfoot distribution throughout Italy, not only in some regions, should be studied and described. From the results, move to the section discussion - speculations and presumptions of the authors. Please expand the discussion. Compare the occurrence of Clubfoot throughout Italy to other countries. Conclusions: please describe the distribution of Clubfoot throughout Italy, not only in some regions. Please correct the styling and punctuation throughout the text.

Author Response

Dear Reviewer,

Thank you for your comment.

We made some of the changes you suggested throughout the manuscript.

We are not able to describe the prevalence of clubfoot throughout Italy as you recommend because we do not have such data.

As we describe in our paper, unfortunately, not all the Regions fill the Cedap Registry.

Te aim of the work is to frame the data we have and we hope it could be a way to encourage all the regions to share their data.

Round 2

Reviewer 1 Report

Dear authors and editor

I would like to justify why I do not recommend that this work be published in its present form.

Results and discussion.

Discussion is very poor.

The results are a simple description of data with little explanation. A poor statistical treatment is performed and other related variables are not considered.

Author Response

As we already explained this is a preliminary study that we would like to improve.

English language has been re examined by a mother tongue reviewer.

Anyway, we thank you for your suggestions.

Reviewer 2 Report

The authors made the suggested changes. Manuscript acceptable.

Author Response

Thank you for the suggestions that helped improve our manuscript.

This manuscript is a resubmission of an earlier submission. The following is a list of the peer review reports and author responses from that submission.

Round 1

Reviewer 1 Report

The authors report data regarding clubfoot in Italy using the Cedap reporting system. They consider current data from EUROCAT "poor" and data for Italy "incomplete". However the EUROCAT registries have Data Quality Indicators to ensure quality. They use common protocols to ensure accuracy of reporting, classification and coding. For example Cedap is only one of multiple ascertainment methods used by The Emila Romagna registry to verify cases and has been considered a Gold Standard. The coverage of Italy for clubfoot data is much greater than quoted (Mantova, Milan, Sicily, Veneto  are all EUROCAT registries ). 

It would be useful to look at ethnic difference that can be derived from CeDAp using mothers citizenship. The English  requires mother tongue revision. The data should be presented in comparison to EUROCAT.  Other sources of data, verification of cases and quality issues could strengthen the paper

Author Response

Thank you for your advice.

It could certainly be very interesting to analyse ethnic differences among the mothers of children with CTEV. On the other hand, according to the data reported in literature, the highest prevalence is reported in Maori, Polynesian and Hawaiian population. These ethnic groups are little or not at all present in Italy. For this reason, we have not considered the data regarding ethnic groups that instead could be a useful tip for further analysis.

The Eurocat registry is an high quality system for the epidemiological surveillance of congenital anomalies, but we considered the data of the italian regions as “poor” because not all the districts have provided full data about new-borns affected by clubfoot; furthermore Italian data given by Eurocat were not available for several years. As an example, Emilia Romagna transmitted the data since 1981, while Campania, Sicily and other regions gave the record on a discontinuous basis.

We decided to use CeDap registry data because it was the most complete public registry of epidemiological and socio demographic information about newborns in Italy.

However, we believe that the records we included in our paper allow to extrapolate regional/national trend.

English revision of the manuscript has been made

Reviewer 2 Report

Dear editors!

It is of great importance to use such registries.

However, I do not understand why only using data between 2013-2017.

Can you also provide the data from 2018-2020?

The problem is you can only guess why the numbers behave differently.

For my opinion there is a need to present a bit more data - maby than it is acceptable for publication.

With kind regards

Author Response

Thank you for your comment.

It was very difficult to find the precise data about clubfoot, since they are not public data and the access to these data is obtained through a complicated procedure requested directly from the Italian Ministry of Health. Furthermore, the records are not updated real time by the Ministry of Health.

The Cedap registry provides only aggregated data that cannot be extrapolated by pathology; they are divided by macro-categories and those categories concerning clubfoot are included in the group of musculoskeletal malformations.

In addition, every Italian region gave the records on a discontinuous basis with missing periods.

Certainly more data would enrich the paper and we will keep this observation in mind for future updates.                                                                                           

English revision of the manuscript has been made.

Reviewer 3 Report

I would like to justify why I do not recommend that this work be published in its present form.

ABSTRACT:

Background: Describing CeDAP is not the background of the study for me.

Keywords: I suggest adding two words:  clubfoot and congenital talipes equinovarus.

Introduction

Line 41. All your cases present congenital clubfoot deformity.  Fewer of them have genetical implications and other children have postural etiopathogenesis. So, the sentence: “The congenital defect differs from postural clubfoot because it is flexible” is very confusing.

Would you mean “other congenital defects as metatarsus varus? 

Line 43-45. “Postural clubfoot is a structurally normal foot and, despite an abnormal resting position, the doctor can correct it with specific physical manipulations. Whereas…foot”.

From my experience, I cannot agree with that comment, even postural clubfoot tends to recur and Achilles tenotomy is almost always recommended. In addition, if the treatment is not early and the appropriate treatment is established, the postural clubfoot presents anatomical and functional alterations.

Materials and Methods

Line 95 “the possible presence of congenital malformations or the causes of neonatal mortality”. Who makes the diagnosis, midwife or obstetrician? I mean, is not the deformation stablished by a paediatrician?

Line 111. Were all newborns, 1379 alives?

Results and discussion.

The results are a simple description of data without any explication. No stadistical treatment or other related variables are performed.

Discussion is very poor.

Round 2

Reviewer 2 Report

Dear Authors!

Thank you for your answers.

I think the English editing process makes the paper better.

However, the missing data can lead to wrong statements, therefore it will be necessary in future to work it up more precisely.

As it should form a basis for future studies my decision is accept.

With kind regards